# Complexation-driven assembly of imine-linked helical receptors showing adaptive folding and temperature-dependent guest selection

Geunmoo Song[1], Seungwon Lee[1] & Kyu-Sung Jeong [1] ✉

The development of synthetic receptors capable of selectively binding guests with diverse structures and multiple functional groups poses a significant challenge. Here, we present the efficient assembly of foldamer-based receptors for monosaccharides, utilising the principles of complexation-induced equilibrium shifting and adaptive folding. Diimine **4** can be quantitatively assembled from smaller components when D-galactose is added as a guest among monosaccharides we examined. During this assembly, dual complexation-induced equilibrium shifts toward both the formation of diimine **4** and the conversion of D-galactose into α-D-galactofuranose are observed. Diimine **6** is quantitatively assembled in the presence of two different guests, methyl β-D-glucopyranoside and methyl β-D-galactopyranoside, resulting in the formation of two dimeric complexes: (**6**-*MP*)$_2$⊃(methyl β-D-glucopyranoside)$_2$ and (**6**-*MM*)$_2$⊃(methyl β-D-galactopyranoside•2H$_2$O)$_2$, respectively. These two complexes exhibit distinct folding structures with domain-swapping cavities depending on the bound guest and temperature. Interestingly, (**6**-*MM*)$_2$⊃(methyl β-D-galactopyranoside•2H$_2$O)$_2$ is exclusively formed at lower temperatures, while (**6**-*MP*)$_2$⊃(methyl β-D-glucopyranoside)$_2$ is only formed at higher temperatures.

Preorganisation and complementarity are two key concepts in the design of synthetic receptors based on the lock and key principle[1,2]. However, it is challenging to design synthetic receptors that have preorganised binding cavities with high complementarity, particularly for flexible guests with multi-functional groups. Often, a certain degree of flexibility is inevitable to facilitate an induced fit[3] between interacting partners, which may potentially result in decreased affinities and selectivities. The optimisation of binding properties can be achieved through iterative modification and evaluation processes. Another strategy involves complexation-driven equilibrium shifting[4–7] to assemble specific receptors that complement to the added guests, utilising dynamic covalent bonds under reversible conditions[8–11]. In this method, receptors can be assembled effectively or quantitatively only when tightly binding guest are present together in reaction mixtures.

Unlike conventional synthetic receptors, foldamer-based receptors[12–15] are capable of creating binding cavities for guests during the folding process. These binding cavities can be modified either by altering the repeating components[16–19] or by changing their folding structures[20]. However, implementing the latter method in the development of synthetic receptors is extremely difficult due to the inherent difficulty of tailoring and predicting the folding structures, particularly higher-order assemblies[21]. We believe that this challenge can be greatly alleviated by employing the principle of complexation-driven equilibrium shifting, which may provide opportunities for

[1]Department of Chemistry, Yonsei University, Seoul 03722, South Korea. ✉e-mail: ksjeong@yonsei.ac.kr

discovering supramolecular receptors that are not accessible through rational design.

In this context, we describe the assemblies and binding properties of foldamer-based receptors, diimines **4** and **6**, which exhibit guest-adaptive folding structures. These diimines can be assembled quantitatively from smaller molecular components only when perfectly fitting guests are present. Diimine **4**, derived from benzene-1,3-diamine, folds into a prototypical single-stranded helix with an internal cavity that only encapsulates α-D-galactofuranose, not the other galactose isomers. Another diimine **6**, derived from 9*H*-fluorene-2,7-diamine, produces two dimeric complexes that display distinct folding structures with domain-swapping[22,23] cavities for guest binding. Interestingly, the bound guest can be completely switched by adjusting the temperature: methyl β-D-glucopyranoside at higher temperatures, but methyl β-D-galactopyranoside at lower temperatures.

## Results

### Design principles

We chose imine bonds[24] as a dynamic linker between small molecular components to construct foldamer-based receptors based on complexation-driven equilibrium shifting. Imine bonds have been widely used in the development of functional supramolecular assemblies[25,26] in dynamic covalent chemistry. These bonds have also been used in the chain elongation of arylene ethynylene foldamers to produce longer sequences with helical structures[27,28]. The formation of imines generally results in a mixture of the desired imine, along with the unreacted aldehyde and amine in a solution. This equilibrium can be completely shifted toward the imine formation through strong binding between a guest and the imine, as depicted in Fig. 1a. In this study, we selected monosaccharides as guests that have multiple hydroxyl groups capable of forming hydrogen bonds with the imine product. Synthetic receptors for carbohydrates have been widely studied[16,29–40], but achieving selective binding of specific carbohydrates remains a great challenge due to the structural and functional group similarity.

Tetramer **1**, as a precursor, consists of two indolocarbazole units and two naphthyridine moieties that are connected alternatively through ethynyl bonds[39] (Fig. 1c). In particular, a naphthyridine-2-carbaldehyde unit is carefully chosen for the imine formation with an aromatic diamine. Previously, we observed a [4 + 2] cycloaddition reaction between adjacent helical turns upon imine formation when an analogous strand with a pyridine-2-carbaldehyde was used[41]. To avoid this reaction, we replaced the pyridine unit with a naphthyridine to deliberately misalign the iminoaryl and ethynyl reaction partners, thereby preventing unwanted [4 + 2] cycloaddition. Tetramer **1** was prepared by repetitive Pd(Ph$_3$P)$_2$Cl$_2$/CuI-catalysed coupling reactions[42] between aryl acetylenes and halides, and the synthetic details are described in the Supplementary Information, together with spectroscopic and physical properties of all new compounds.

### Assembly of diimine 4

Through the screening process, we found that diimine **4** was efficiently formed from tetramer **1** (2 equiv.) and benzene-1,3-diamine **2** (1 equiv.) in the presence of chloroacetic acid (0.2 equiv.) as the acid catalyst in 2% (v/v) DMSO-$d_6$/CD$_2$Cl$_2$ at 39 ± 1 °C. In the absence of any guest, the reaction yielded a mixture of unreacted tetramer **1** (29%), monoimine **3** (28%), and diimine **4** (43%) (Fig. 1f). Subsequently, this reaction was examined in the presence of various monosaccharide guests (3 equiv.). All the reactions resulted in broad, complicated $^1$H NMR spectra, but the presence of D-galactose led to a well-resolved $^1$H NMR spectrum that corresponded to a complex between diimine **4** and D-galactose (Fig. 1f and Supplementary Fig. 1). The desired diimine **4** was formed quantitatively (~ 97% isolated yield), and its complex with D-galactose was sufficiently stable to exhibit a well-resolved $^1$H NMR spectrum under the given conditions. Neither monoimine **3** nor unreacted

tetramer **1** was detected in the spectrum, which was also confirmed by HPLC analyses (Supplementary Table 1 and Fig. 4).

The $^1$H NMR spectrum indicated that the bound isomer of D-galactose was only α-D-galactofuranose (Supplementary Fig. 8). D-Galactose is known to exist as five structural isomers[43] and the relative distributions depend on the environment. In this study, the distributions were determined to be 48% for α-D-galactopyranose (α-D-GP), 19% for β-D-galactopyranose (β-D-GP), 15% for α-D-galactofuranose (α-D-GF), 18% for β-D-galactofuranose (β-D-GF), and <1% (not detected) for an acyclic form when D-galactose was dissolved in 10% (v/v) DMSO-$d_6$/CD$_2$Cl$_2$ and allowed to stand for 24 h at room temperature[16] (Supplementary Fig. 7). It should be noted that diimine **4** was also formed quantitatively in the presence of only 1 equiv. of D-galactose, indicating that all isomers were converted into α-D-GF during the assembly reaction (Supplementary Fig. 2). Furthermore, $^1$H NMR and circular dichroism (CD) spectra demonstrate that D-galactose was completely converted into α-D-GF within 24 h at room temperature when diimine **4** (1 equiv.) was present in 10% (v/v) (deuterated) DMSO/CH$_2$Cl$_2$ (Fig. 2a and Supplementary Fig. 15). These observations indicate that complex formation between diimine **4** and α-D-GF induces dual equilibrium shifts in both the receptor and the guest for optimal complementarity[40]. The association constant between diimine **4** and α-D-GF was determined by CD titration in 10% (v/v) DMSO/CH$_2$Cl$_2$ (containing 0.04–0.06% water) at 25 ± 1 °C. Thirteen separate stock solutions, each with different molar ratios of **4** and D-galactose, were prepared and allowed to stand at room temperature for 24 h before the CD measurement to ensure complete isomerisation. Non-linear regression analysis[44] afforded an observed association constant ($K_{obs}$) of 5.40 ( ± 0.17) × 10$^4$ M$^{-1}$ (Supplementary Fig. 20). It is noted that this value is not the intrinsic binding constant between diimine **4** and α-D-GF but the observed one, calculated without considering the structural isomer distributions of D-galactose. Selective binding of **4** with α-D-GF was confirmed by competitive binding experiments using mixed monosaccharide guests including D-glucose, D-galactose, D-mannose, D-fructose, and D-arabinose. Three separate solutions were prepared, each containing **4** and different combinations of mixed guests (3 equiv. of each guest) in 5% (v/v) DMSO-$d_6$/CD$_2$Cl$_2$. A well-resolved, sharp $^1$H NMR spectrum was exclusively observed in the solution containing D-galactose among the mixed guests. Notably, this spectrum was identical to the one obtained for the complex between **4** and D-galactose as a single guest (Fig. 1f). In contrast, the other solutions lacking D-galactose in the mixed guests yielded broad spectra (Supplementary Fig. 16). These results indicate that α-D-GF selectively and strongly binds to **4**, enabling the quantitative assembly of **4** under the given conditions. For comparison, CD titrations for D-glucose, D-mannose, D-fructose, and D-arabinose were also performed, and the observed association constants were estimated based on 1:1 binding modes. As anticipated, all these values are smaller by more than one order of magnitude compared with that ($K_{obs}$ = 5.40 × 10$^4$ M$^{-1}$) of D-galactose (Supplementary Table 6 and Figs. 21–23).

The structure of complex **4**⊃α-D-GF • H$_2$O was unambiguously determined by single crystal X-ray diffraction. Single crystals were obtained by vapor diffusion of pentane into a chloroform solution containing an approximately 1:1 molar ratio of **4** and D-galactose. As shown in Fig. 2d, diimine **4** folds into a helical structure with an internal tubular cavity, in which α-D-GF, an isomeric form of D-galactose, is encapsulated with one water molecule. The complex **4**⊃α-D-GF • H$_2$O is stabilised by 14 hydrogen bonds between **4**, α-D-GF and a water molecule, along with π-stacking interactions between helical turns[45] (Fig. 2e). Computer modeling[46,47], utilising this crystal structure, indicates that the two hydrogen bonds, initially formed with the anomeric OH, are disrupted when substituting α-D-GF with β-D-GF. In the case of the pyranose isomers, the hydrogen-bonding array of **4**⊃α-D-GF • H$_2$O are significantly perturbed, resulting in even fewer hydrogen bonds between **4** and the pyranose isomers (Supplementary Fig. 41).

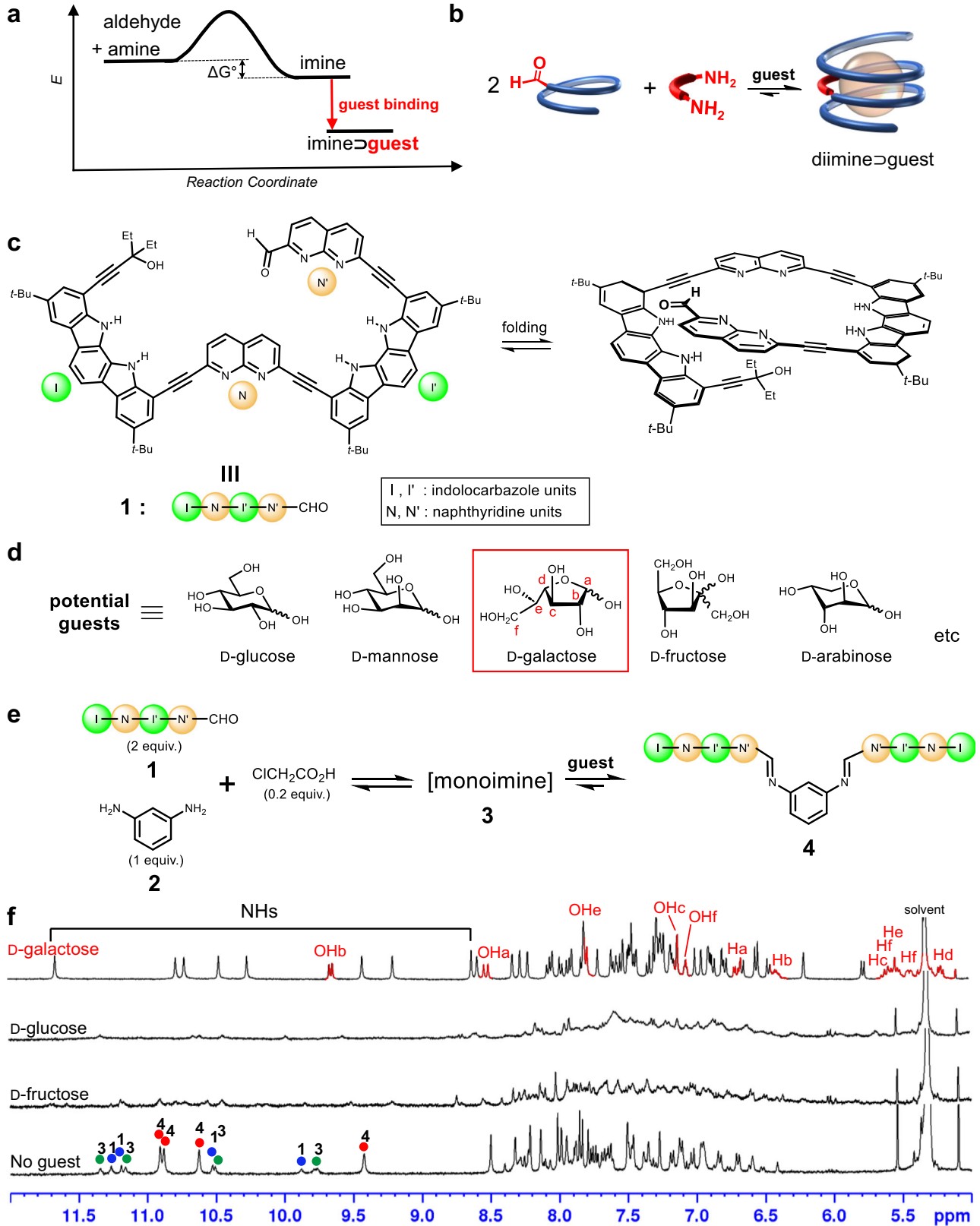

**Fig. 1 | Principle of imine assembly, molecular structures, and ¹H NMR spectra of reaction mixtures. a** Hypothetical energy profile for a complexation-driven equilibrium shift toward imine formation. **b** Schematic representation of a complexation-induced equilibrium shift. Chemical structures of **c** tetramer **1** and **d** monosaccharides used as potential guests. **e** Assembly of diimine **4**. **f** Partial ¹H NMR (400 MHz, 25 °C) spectra of reaction mixtures of **1** (2 equiv.), benzene-1,3- diamine **2** (1 equiv.), chloroacetic acid (0.2 equiv.), and each guest (3 equiv.) in 2% (v/v) DMSO-$d_6$/CD$_2$Cl$_2$ after 24 h heating at 39 °C in rubber septum-sealed NMR tubes. Signals of α-D-galactofuranose are highlighted in red in the top spectrum. In the absence of a guest (bottom spectrum), the NH signals of tetramer **1**, monoimine **3**, and diimine **4** are marked with blue, green, and red circles, respectively.

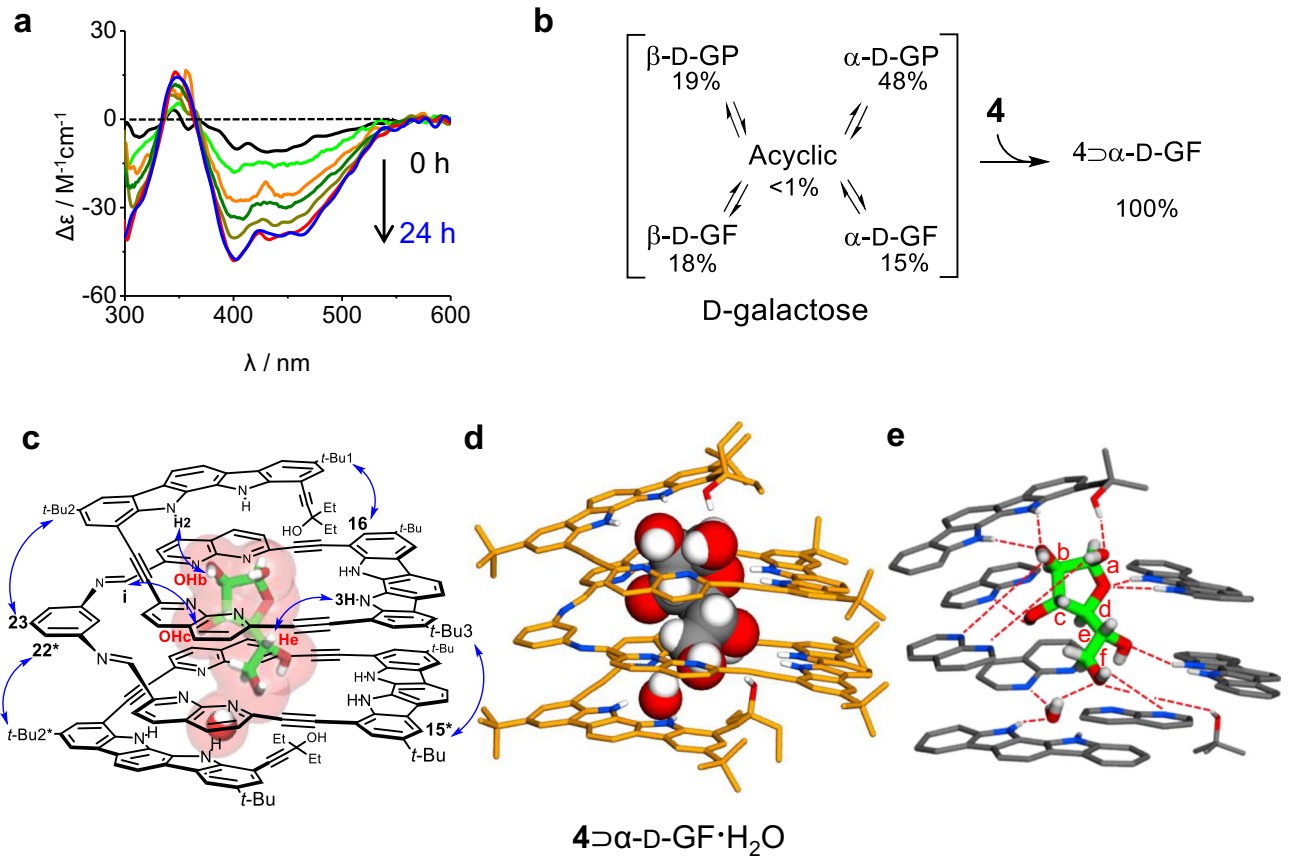

**Fig. 2 | Isomerisation of D-galactose and X-ray structures of complex 4⊃α-D-GF • H₂O. a** Time-dependent CD spectra of **4** (2.00 × 10⁻⁵ M, 25 ± 1°C) upon the addition of D-galactose (1 equiv.) in 10% (v/v) DMSO/CH₂Cl₂. The CD intensities are gradually increased until 12 h (red) and then remain constant. **b** Relative distributions of the five D-galactose isomers in 10% (v/v) DMSO-$d_6$/CD₂Cl₂ and their conversions to α-D-GF upon complexation. GF: galactofuranose. GP: galactopyranose. **c** Molecular structure of complex **4⊃α-D-GF • H₂O** with NOE correlations. **d** X-ray crystal structure of **4⊃α-D-GF • H₂O**, where diimine **4** and α-D-GF • H₂O are shown in tube and CPK representation, respectively. **e** Hydrogen bonds between **4**, α-D-GF, and a water molecule determined from the X-ray crystal structure.

The ¹H NMR and CD studies are consistent with the X-ray crystal structure. Upon binding of **4** with α-D-GF, the four NH signals split to eight signals between 11.7 and 8.6 ppm, and the aromatic CH signals are upfield-shifted up to Δδ = 1.1 ppm. The OH signals of bound α-D-GF appear between 9.7 and 7.0 ppm (Fig. 1f). Furthermore, the 2D-ROESY spectrum shows characteristic NOE correlations between the non-adjacent hydrogen atoms of **4** (*t*-Bu1-H16, *t*-Bu2-H23, *t*-Bu2*-H22*, and *t*-Bu3-H15*), as well as intermolecular NOE cross peaks between **4** and α-D-GF (NH2-OHb, Hi-OHc, and NH3-He). (Fig. 2c and Supplementary Fig. 9). These results are in good agreement with the crystal structure of **4⊃α-D-GF • H₂O**. As shown in Fig. 2d, diimine **4** folds into a left-handed helix with approximately three turns. Consistent with this structure, diimine **4** exhibited strong CD signals when complexed with D-galactose. The binding of its enantiomeric L-galactose resulted in a symmetrical CD spectrum with the opposite Cotton effect, indicating that the helix orientation depends on the guest chirality (Supplementary Fig. 19).

## Assembly of diimine 6

To prepare another foldamer-based receptor with imine linkages, we replaced the aromatic diamine with 9*H*-fluorene-2,7-diamine **5** in which the two amino groups are further apart and more divergent. We conducted the coupling reaction between tetramer **1** (2 equiv.) and 9*H*-fluorene-2,7-diamine **5** (1 equiv.) in the presence of various monosaccharides (Fig. 3a and Supplementary Fig. 3). Among them, we only

observed well-resolved ¹H NMR signals in the presence of two guests, methyl β-D-glucopyranoside (me-β-D-glc) and methyl β-D-galactopyranoside (me-β-D-gal). Diimine **6** was assembled quantitatively in both reactions, and its complexes with the added guests were sufficiently stable to display well-resolved ¹H NMR spectra under the given conditions.

First, the structure of the complex formed between **6** and me-β-D-glc was determined using single-crystal X-ray diffraction (Fig. 3c). Single crystals were obtained at room temperature by vapor diffusion of pentane into a 1,2-dichloroethane solution containing a mixture obtained from the coupling reaction. The complex was found to be (**6**-*MP*)₂⊃(me-β-D-glc)₂, existing in a dimeric form with two identical cavities for binding me-β-D-glc. Interestingly, these two cavities are formed via domain swapping, reminiscent of protein folds[48,49]. When dimerised, the two strands of diimine **6** exchange helical elements and produce two identical cavities of helices. Specifically, each diimine strand folds into two separate helices with partial cavities; one side folds into an *M*-helix, whereas the other forms a *P*-helix (Fig. 3b). In the *M*-helix, all four repeating monomers are fully folded. However, the *P*-helix is partially unfolded; the naphthyridine monomer next to the imino linkage is rotated by approximately 180°, causing its two nitrogen atoms to be placed outside the helical backbone. Hence, the two helical components with opposite orientations (*M* and *P*) are placed on the same side of the fluorene plane in a cisoid geometry, but no mirror plane exists in the middle of the strand. The two strands of diimine **6**

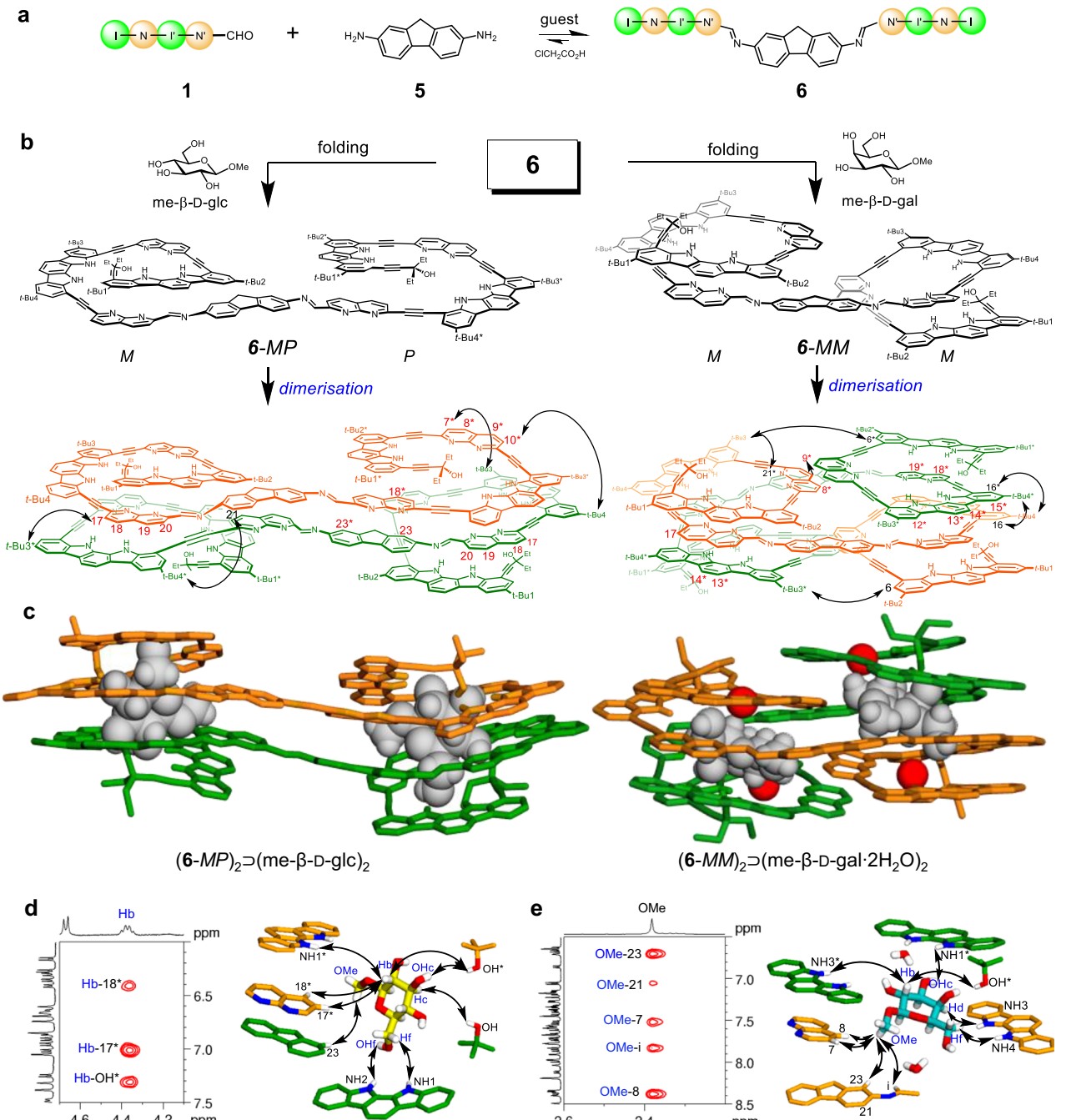

**Fig. 3 | Assembly and structural analysis of (6-MP)₂⊃(me-β-D-glc)₂ and (6-MM)₂⊃(me-β-D-gal • 2H₂O)₂.** a Assembly of diimine **6**. b Folding structures of **6** and its dimers observed in X-ray structures. c X-ray crystal structures of (**6**-MP)₂⊃(me-β-D-glc)₂ and (**6**-MM)₂⊃(me-β-D-gal•2H₂O)₂. me-β-D-glc: methyl β-D-glucopyranoside. me-β-D-gal: methyl β-D-galactopyranoside. Two separate strands are shown in orange and green tubes, and guests and H₂O are shown in grey and red CPK's, respectively. All CH hydrogen atoms and t-Bu groups in **6** are omitted for clarity. d Selected ¹H-¹H ROESY spectrum (left) and NOE correlations between **6** and me-β-D-glc (right). e Selected ¹H-¹H ROESY spectrum (left) and NOE correlations between **6** and me-β-D-gal (right).

become dimerised via face-to-face stacking in an antiparallel manner. In other words, the M-helix of one strand stacks precisely on the P-helix of another, and vice versa (Fig. 3b). Consequently, two identical cavities are generated with a large aryl contact between the two diimine strands. Each cavity can accommodate one molecule of me-β-D-glc by forming 9 hydrogen bonds (Supplementary Table 16 and Fig. 36), which results in a 2:2 (diimine/guest) complex, (**6**-MP)₂⊃(me-β-D-glc)₂.

The crystal structure of (**6**-MP)₂⊃(me-β-D-glc)₂ is consistent with its ¹H NMR spectra. When **6** and me-β-D-glc were mixed in a 1:1 ratio, all four OH signals of me-β-D-glc were largely downfield-shifted between

11.0 and 7.7 ppm, as a result of hydrogen bonding. Eight NH signals of **6** appear between 11.8 and 8.5 ppm, as expected upon binding of a chiral guest me-β-D-glc (Supplementary Fig. 11). The CH signals of naphthyridine (H7*–H10*, H18* and H17–H20) and fluorene (CH23, CH23*) are remarkably upfield-shifted (Δδ = 0.6–1.8 ppm) due to tight stacking resulting from the dimerisation of two diimine strands (Fig. 3b). Furthermore, the methyl signal of me-β-D-glc shifted upfield from 3.4 to 1.4 ppm due to the shielding effect of adjacent aryl planes (Supplementary Table 4 and Fig. 35). Finally, the 2D-ROESY spectrum shows NOE correlations between the two different diimine strands (Fig. 3b) and also

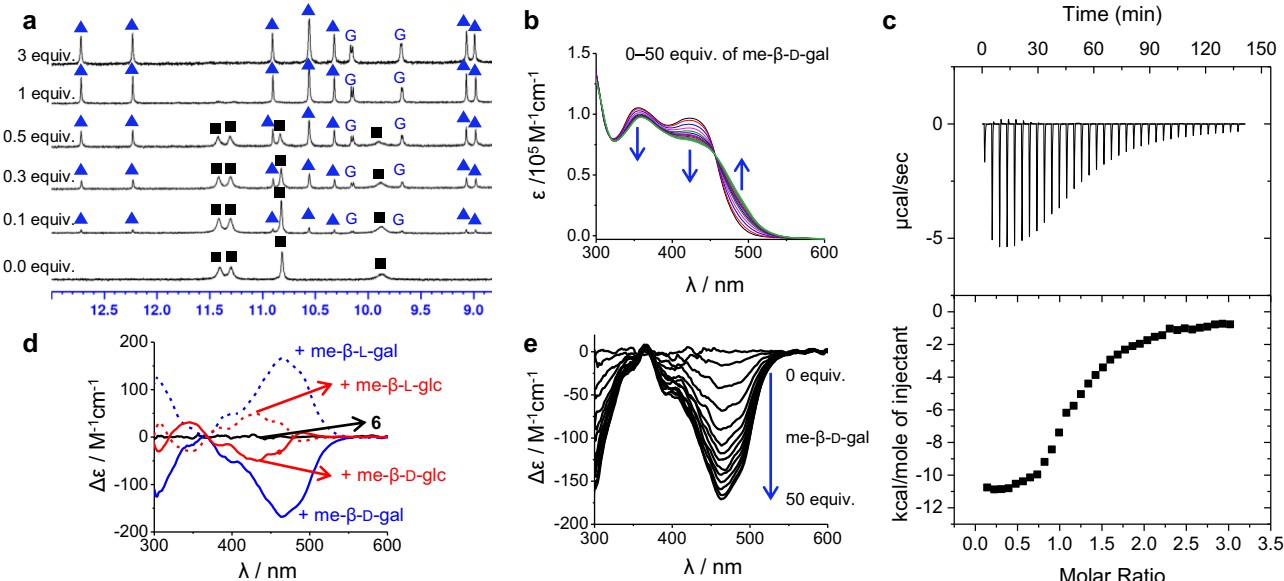

**Fig. 4 | Binding studies of diimine 6 with me-β-ᴅ-gal. a** Partial ¹H NMR (400 MHz) spectra of **6** (1.50 × 10⁻³ M) with increasing me-β-ᴅ-gal in 5% (v/v) DMSO-$d_6$/CD₂Cl₂ at 25 °C. The NH signals of unbound **6** and its 2:2 complex are marked with black squares and blue triangles, respectively. The signals of bound me-β-ᴅ-gal are marked as G. **b** UV-visible spectral changes of **6** (2.00 × 10⁻⁵ M, 25 °C) upon titrating with me-β-ᴅ-gal in 5% (v/v) DMSO/CH₂Cl₂. **c** Experimental ITC results of **6** (1.00 × 10⁻⁴ M) with me-β-ᴅ-gal in 5% (v/v) DMSO/CH₂Cl₂ at 22 °C. **d** CD spectra of **6** (2.00 × 10⁻⁵ M, 25 °C) in the absence and presence of excess guests (~100 equiv.) in 5% (v/v) DMSO/CH₂Cl₂. **e** CD spectral changes of **6** upon titrating with me-β-ᴅ-gal in 5% (v/v) DMSO/CH₂Cl₂.

between diimine **6** and me-β-ᴅ-glc (Fig. 3d). Notably, NOE cross peaks were observed between the rotated naphthyridine CH protons (H17*, H18*) and the Hb proton of me-β-ᴅ-glc. These data are in good agreement with the X-ray structure of (**6**-*MP*)₂⊃(me-β-ᴅ-glc)₂.

To determine the X-ray structure of the complex between diimine **6** and me-β-ᴅ-gal, we initially attempted to grow single crystals of the complex under various conditions, but all attempts were unsuccessful. Racemic crystallization[50] was then conducted using equal amounts of **6**, me-β-ᴅ-gal, and its enantiomer me-β-ʟ-gal. Fortunately, single crystals suitable for X-ray diffraction were obtained, with the complexes identified as (**6**-*MM*)₂⊃(me-β-ᴅ-gal•2H₂O)₂ (Fig. 3c) and (**6**-*PP*)₂⊃(me-β-ʟ-gal•2H₂O)₂ (Supplementary Fig. 38). Interestingly, the folding structure of diimine **6** was considerably different from that of the previous complex (**6**-*MP*)₂⊃(me-β-ᴅ-glc)₂. Each diimine strand folds into a helical structure containing two identical half cavities, resulting in twofold symmetry. Both half cavities are left-handed helices (*M, M*) when complexed with me-β-ᴅ-gal, whereas they are right-handed helices (*P, P*) with me-β-ʟ-gal. In addition, the two half cavities are positioned in a transoid geometry around the fluorene plane. Two strands of **6** are assembled to afford a dimeric receptor with two identical tubular cavities capable of binding guest molecules. Domain-swapping cavities are formed by precisely stacking half of the cavity of one strand on the top or bottom of the other strand (Fig. 3b). The resulting cavities (~298 Å³) for me-β-ᴅ-gal are somewhat larger compared to the previous cavities (~210 Å³) for me-β-ᴅ-glc (Supplementary Fig. 40). The larger cavity is possibly responsible for two water molecules binding together with me-β-ᴅ-gal in each cavity to form (**6**-*MM*)₂⊃(me-β-ᴅ-gal•2H₂O)₂. The X-ray structure of (**6**-*MM*)₂⊃(me-β-ᴅ-gal•2H₂O)₂ was confirmed by ¹H NMR studies in 2% (v/v) DMSO-$d_6$/CD₂Cl₂ (containing ~0.06% H₂O) (Supplementary Fig. 13). When **6** and me-β-ᴅ-gal were mixed, eight distinct NH signals of **6** were observed between 12.8 and 8.5 ppm. The CH signals for the indolocarbazole (H12*–H15*) and naphthyridine (H8*, H9*, H18*, and H19*) were significantly upfield-shifted ($\Delta\delta$ = 0.4–1.7 ppm) due to dimerisation of the two diimine strands (Fig. 3b). All OH signals of me-β-ᴅ-gal were largely downfield-shifted ($\delta$ = 10.2–6.0 ppm) due to the hydrogen bonding formation. Furthermore, the 2D-ROESY spectrum displays NOE correlations between the two different diimine strands (Fig. 3b) and also

between diimine **6** and me-β-ᴅ-gal (Fig. 3e), which are fully consistent with the X-ray structure.

## Determination of association constants

The binding properties of diimine **6** with me-β-ᴅ-glc and me-β-ᴅ-gal were investigated using various techniques, including ¹H nuclear magnetic resonance (NMR), circular dichroism (CD), ultraviolet-visible spectroscopy, and isothermal titration calorimetry (ITC). Firstly, we conducted ¹H NMR studies to reveal complex formation between diimine **6** and me-β-ᴅ-glc or me-β-ᴅ-gal. When me-β-ᴅ-glc or me-β-ᴅ-gal was added to a solution of **6** in 5% (v/v) DMSO-$d_6$/CD₂Cl₂ (containing 0.04−0.06% water) at 25 °C, a new set of ¹H NMR signals was observed due to the slow exchange between unbound species and the complex (Fig. 4a and Supplementary Fig. 24). Specifically, ¹H NMR signals corresponding to a 2:2 (**6**/guest) complex were only observed, even in the presence of small amounts (<0.3 equiv.) of the guest. As the guest concentration increased, the signals for the 2:2 complex were intensified at the expense of the signals for unbound **6**. No other signals appeared throughout the titrations. These results indicate that possible intermediates, such as 1:1, 2:1, and 1:2 (**6**/guest) complexes, are too unstable to be observed under the given conditions. Therefore, we determined the association constants ($K$, M⁻³) for the formation of the 2:2 complexes as a single-step process. This assumption is also consistent with UV-visible titrations and ITC experiments (*vide infra*).

Secondly, diimine **6** was CD-inactive by itself, but it displayed characteristic CD signals when complexed with chiral guests such as methyl glycosides. For instance, binding of me-β-ᴅ-gal and me-β-ᴅ-glc led to induced CD signals with negative Cotton effects, and their enantiomers, me-β-ʟ-gal and me-β-ʟ-glc, gave exactly opposite Cotton effects (Fig. 4d). Notably, the CD intensities were much stronger when binding me-β-gal compared to binding me-β-glc. These observations are consistent with the X-ray crystal structures of the two complexes. Diimine **6** adopts helical conformations with the same orientation (*M,M*) in (**6**-*MM*)₂⊃(me-β-ᴅ-gal•2H₂O)₂, but it folds into a pseudo-meso structure (*M,P*) in (**6**-*MP*)₂⊃(me-β-ᴅ-glc)₂, thereby offsetting the ellipticity. CD titration experiments (Fig. 4e) yielded an association constant of log $K$ = 12.9 (±0.1) between **6** and me-β-ᴅ-gal in 5% (v/v) DMSO/CH₂Cl₂

**Table 1 | Titration results between 6 and me-β-D-glc, or me-β-D-gal in 5% (v/v) DMSO/CH₂Cl₂ (containing 0.04–0.06% water)**

| Entry | Guest | Log K (2:2 complex) | | | ΔG° | ΔH° | TΔS° |
|---|---|---|---|---|---|---|---|
| | | CD[a] | UV-visible[a] | ITC[b] | | (kJ mol⁻¹) | |
| 1 | me-β-D-glc | ND[c] | 13.5 ± 0.1 | 13.5 ± 0.1 | −76.4 | −56.0 | +20.4 |
| 2 | me-β-D-gal | 12.9 ± 0.1 | 13.0 ± 0.1 | 13.2 ± 0.1 | −74.4 | −130.2 | −55.8 |

$K = [\text{diimine}_2 \cdot \text{guest}_2]/[\text{diimine}]^2[\text{guest}]^2$. All titrations were duplicated at [a] 25 ± 1 °C and [b] 22 ± 1 °C. [c] Not determined.

(containing 0.04–0.06% water) at 25 ± 1 °C, which was calculated assuming a one-step 2:2 binding mode using ReactLab EQUILIBRIA 1.1[51]. Thirdly, UV-visible titrations showed isosbestic points at 390, 433, and 446 nm for me-β-D-glc and 455 nm for me-β-D-gal, which did not change during titrations (Fig. 4b and Supplementary Fig. 27). These observations further support the idea of a single-step equilibrium for the formation of the 2:2 complexes, as mentioned earlier in the ¹H NMR spectra. The association constants were calculated as log K = 13.5 (± 0.1) for me-β-D-glc and log K = 13.0 (± 0.1) for me-β-D-gal.

Finally, thermodynamic parameters (ΔH°, ΔS°) for binding were determined using ITC experiments (5% (v/v) DMSO/CH₂Cl₂ containing 0.04–0.06% water, 22 ± 1 °C). (Fig. 4c and Supplementary Fig. 29). Each binding isotherm showed a sigmoidal curve with a single inflection point at the molar ratio (6/guest) of approximately 1. This is also in agreement to the single-step formation of the 2:2 complexes as described in the ¹H NMR and UV-visible experiments[52]. The titration curves were analysed using HypCal software[53,54], and the enthalpy (ΔH°) and entropy values (TΔS°) for the binding of 6 with me-β-D-glc were −56.0 kJ mol⁻¹ and +20.4 kJ mol⁻¹, respectively. This result indicates that the formation of complex (6-MP)₂⊃(me-β-D-glc)₂ is favourable both enthalpically and entropically under the given conditions. Presumably, diimine 6 contains several water molecules in its cavity under the given conditions (containing 0.04–0.06% water). Binding of me-β-D-glc should release these water molecules, thereby making the binding process entropically favourable. This rationale also supports the relatively small net gain of enthalpy in the binding process, considering that 9 hydrogen bonds are formed between 6 and me-β-D-glc in each cavity. On the other hand, the thermodynamic parameters for the binding of 6 with me-β-D-gal were calculated to be ΔH° = −130.2 kJ mol⁻¹ and TΔS° = −55.8 kJ mol⁻¹, which differ significantly from those for me-β-D-glc binding. These results align with the X-ray structures of the two complexes described earlier. Diimine 6 is fully folded to generate larger binding cavities (-298 Å³) in the X-ray structure of (6-MM)₂⊃(me-β-D-gal•2H₂O)₂. However, it was partially unfolded, resulting in smaller cavities (-210 Å³) in (6-MP)₂⊃(me-β-D-glc)₂. Consequently, two water molecules in each cavity are bound with me-β-D-gal, one on the top and one on the bottom of the guest. As a result, (6-MM)₂⊃(me-β-D-gal•2H₂O)₂ is stabilised by a total of 24 hydrogen bonds, while (6-MP)₂⊃(me-β-D-glc)₂ is stabilised by only 18 hydrogen bonds. These structural features suggest that the binding of me-β-D-gal is favoured enthalpically but disfavoured entropically compared with the binding of me-β-D-glc. As summarized in Table 1, the association constants determined by three different methods are all identical within the error ranges.

To further demonstrate the selective binding of 6 towards me-β-D-glc and me-β-D-gal, we conducted competitive binding experiments. Five separate solutions were prepared in 5% (v/v) DMSO-d₆/CD₂Cl₂, each containing 6 and different combinations of mixed glycoside guests (me-β-D-glc, me-α-D-gal, me-β-D-gal, me-α-D-man, and me-β-D-xyl). Well-resolved ¹H NMR spectra were observed only when the guest mixtures contained me-β-D-glc and/or me-β-D-gal, while other combinations of the guests resulted in broad spectra (Supplementary Fig. 17). These results strongly suggest that both me-β-D-glc and me-β-D-gal bind more strongly to 6 than other methyl glycosides under the given conditions. As anticipated, isothermal titration calorimetry (ITC) experiments demonstrated that the association constants of 6 for me-

α-D-gal, me-α-D-man, and me-β-D-xyl are much smaller than those of me-β-D-glc and me-β-D-gal (Supplementary Table 7 and Figs. 31–33).

## Temperature-controlled guest selection

The binding affinities of diimine 6 with two guests, me-β-D-glc and me-β-D-gal, are comparable to each other at room temperature although the folding structures and binding parameters of the two complexes are significantly different. When 6 was mixed with a 1:1 molar ratio of me-β-D-glc and me-β-D-gal in 5% (v/v) DMSO-d₆/(CD₂Cl)₂ (containing 0.04–0.06% water), two separate sets of ¹H NMR signals were observed (Fig. 5b). These signals corresponded to two complexes, (6-MP)₂⊃(me-β-D-glc)₂ and (6-MM)₂⊃(me-β-D-gal•2H₂O)₂, and their relative intensities were nearly equal at 20 ± 1 °C, as expected based on their comparable stabilities. When the temperature decreased, the intensities of ¹H NMR signals for (6-MM)₂⊃(me-β-D-gal•2H₂O)₂ increased gradually and was exclusively seen at temperatures below −20 °C (Fig. 5b). On the contrary, the signals for (6-MP)₂⊃(me-β-D-glc)₂ became more intense with increasing temperature and were only observed at temperatures above 60 °C. Namely, the enthalpically favourable (6-MM)₂⊃(me-β-D-gal•2H₂O)₂ complex was exclusively formed at lower temperatures, while the entropically favourable (6-MP)₂⊃(me-β-D-glc)₂ complex was only obtained at higher temperatures. This temperature-controlled selection of bound guests[55] may be a unique characteristic of foldamer-based receptors with guest-adaptive folding features.

## Discussion

In conclusion, most synthetic receptors have been prepared by focusing on the covalent arrangement of specific binding cavities for target guests. The binding affinity and selectivity can be further improved through iterative modifications and evaluations[16,17]. Alternatively, synthetic receptors can be prepared using the principle of complexation-induced equilibrium shifting in the presence of the appropriate guests. In this study, we demonstrated that the latter principle is particularly useful in the synthesis of foldamer-based receptors with guest-binding cavities, which are produced through adaptive folding in response to the environment. This approach allows for the discovery of foldamer-based dimeric receptors with domain-swapping cavities, which may not be accessible through rational design. Remarkably, this dimeric receptor completely switches its binding partner with changing temperatures, which involves the reorganisation of the binding cavity through unfolding and refolding processes. By combining guest-adaptive folding with dynamic covalent chemistry[56,57], this study offers a new approach to finding an unpredictable assembly that functions as a synthetic receptor with high affinity and selectivity.

## Methods

### Assembly of diimine 4

Stock solutions of 1 (2.00 × 10⁻³ M, 1 % (v/v) DMSO-d₆/CD₂Cl₂), benzene-1,3-diamine 2 (2.00 × 10⁻² M, CD₂Cl₂), chloroacetic acid (1.00 × 10⁻² M, 0.5 % (v/v) DMSO-d₆/CD₂Cl₂), and a monosaccharide guest (0.2 M, DMSO-d₆) were separately prepared. Using these stock solutions, 400 μL of 1 (2 equiv., 0.8 μmol), 20 μL of benzene-1,3-diamine 2 (1 equiv., 0.4 μmol), 8 μL of chloroacetic acid (0.2 equiv., 0.08 μmol), and 6 μL of a monosaccharide guest (3 equiv., 1.2 μmol) were added to a NMR tube. Subsequently, additional 70 μL of CD₂Cl₂ was added, and

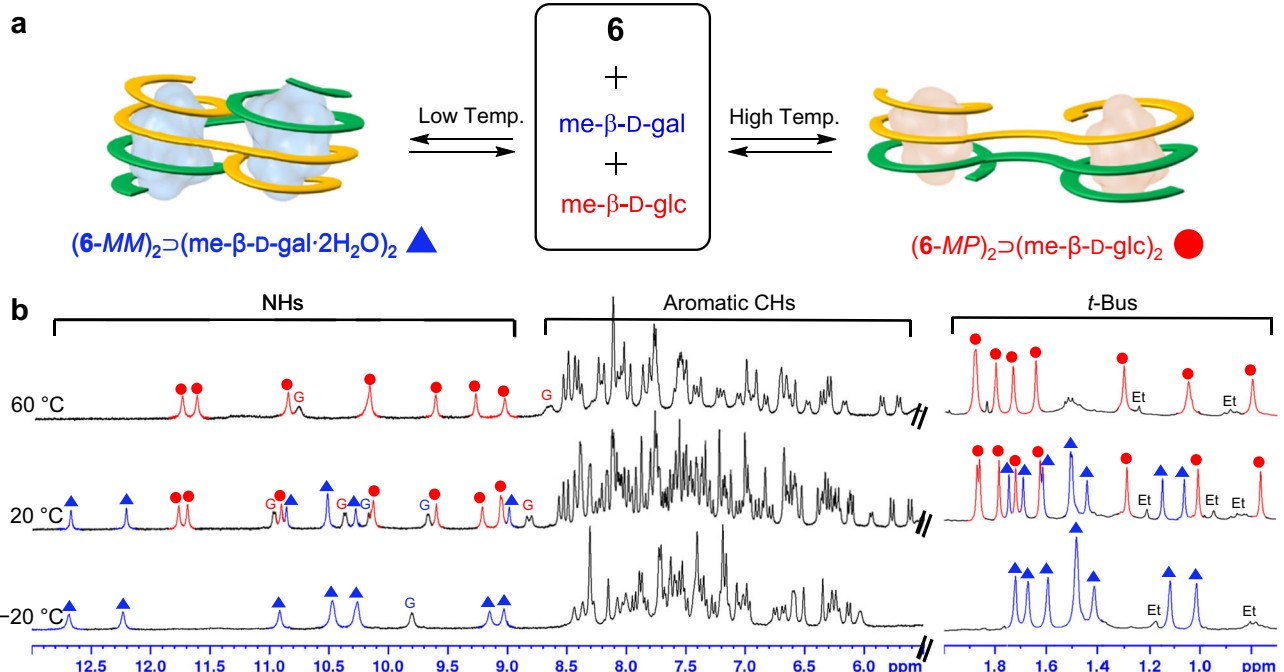

**Fig. 5 | Temperature-dependent $^1$H NMR spectra of diimine 6 in the presence of both me-β-D-glc and me-β-D-gal. a** Temperature-dependent formation of the two different complexes of **6** with me-β-D-glc and me-β-D-gal. **b** Partial $^1$H NMR (300 MHz) spectra of a solution containing **6** ($1.00 \times 10^{-3}$ M), me-β-D-glc (3 equiv.), and me-β-D-gal (3 equiv.) in 5% (v/v) DMSO-$d_6$/(CD$_2$Cl)$_2$ at −20 °C, +20 °C, and +60 °C. $^1$H NMR signals for **(6-MP)$_2$⊃(me-β-D-glc)$_2$** can only be seen at +60 °C, while those for **(6-MM)$_2$⊃(me-β-D-gal•2H$_2$O)$_2$** can be only observed at −20 °C. The signal intensities of the two complexes are nearly equal at +20 °C. The NH and *t*-Bu signals of **(6-MP)$_2$⊃(me-β-D-glc)$_2$** and **(6-MM)$_2$⊃(me-β-D-gal•2H$_2$O)$_2$** are marked with red circles and blue triangles, respectively. The signals for the ethyl groups at the end of diimine **6** are marked as Et's.

the tube was tightly sealed with a rubber septum. Then, the reaction mixture was heated at 39 ± 1 °C for 24 h. Eight monosaccharides were tested as potential guests. $^1$H NMR spectra were recorded and analysed to reveal the best guest for diimine **4** (Supplementary Fig. 1). This guest was used to prepare **4** on a larger scale as follows: **1** (25.3 mg, 2 equiv.), D-galactose (1.9 mg, 1 equiv.), benzene-1,3-diamine **2** (1.1 mg, 1 equiv.), and chloroacetic acid (0.2 mg, 0.2 equiv.) were dissolved in degassed DMSO (0.2 mL) and CH$_2$Cl$_2$ (6.8 mL). The reaction mixture was stirred at 39 ± 1 °C for 24 h in a vial tightly sealed with a septum. The reaction mixture was washed with saturated aqueous NaHCO$_3$ and then concentrated. The residue was filtered through short silica-gel (hexanes: ethyl acetate (EtOAc): methanol: triethylamine (TEA) = 10:10:1:1 (v/v/v/v)) to give **4** (25.4 mg, 97 %) as an orange solid.

### Assembly of diimine 6

The procedure is the same as that for the assembly of **4** except that 9H-fluorene-2,7-diamine **5** was used instead of benzene-1,3-diamine **2**. Fourteen monosaccharides were tested as potential guests (Supplementary Fig. 3). Diimine **6** was prepared on a larger scale as follows: **1** (35 mg, 2 equiv.), methyl β-D-glucopyranoside (8.4 mg, 3 equiv.) (or methyl β-D-galactopyranoside), 9H-fluorene-2,7-diamine **5** (2.8 mg, 1 equiv.) and chloroacetic acid (0.28 mg, 0.2 equiv.) were dissolved in degassed DMSO (0.2 mL) and CH$_2$Cl$_2$ (9.5 mL). The reaction mixture was stirred at 39 ± 1 °C for 24 h in a vial tightly sealed with a septum. The reaction mixture was washed with saturated aqueous NaHCO$_3$ and then concentrated. The residue was then filtered through short silica gel (hexanes : tetrahydrofuran (THF) : methanol : trimethylamine (TEA) = 10:10:1:1 (v/v/v/v)) to give **6** (35.5 mg, 95%) as an orange solid.

### CD titrations

CD titrations were performed using JASCO J-815 spectrometer under the following conditions: scan rate: 500 nm•min$^{-1}$, response time: 1.0 sec, band width: 1.0 nm, accumulations: 2 scans, 25 ± 1 °C. Stock solutions of **4** ($2.20 \times 10^{-5}$ M, CH$_2$Cl$_2$) and guests (2.00–20.0 mM, DMSO) were separately prepared at room temperature. Using these stock solutions, 13 separate solutions (10% (v/v) DMSO/CH$_2$Cl$_2$, containing 0.04–0.06% water) with different molar ratios of **4** and guests were prepared. The same concentration of **4** ($2.00 \times 10^{-5}$ M) was used in all solutions. The solutions were allowed to stand for 24 h at room temperature, after which the CD spectrum of each solution was recorded. The association constants ($K_{obs}$, M$^{-1}$) were determined using Bindfit software[44]. On the other hand, a stock solution of **6** ($2.00 \times 10^{-5}$ M, 5% (v/v) DMSO/CH$_2$Cl$_2$, water content (v/v) 0.04–0.06%) was prepared. Using parts of this solution as a solvent, the stock solution of me-β-D-gal ($3.00 \times 10^{-3}$ M) was prepared. Aliquots of the guest solution were added to the cell containing a **6** solution (2.00 mL). CD spectra were recorded and the association constant (K, M$^{-3}$) was determined using ReactLab software[51].

### UV-visible titrations

UV titrations were performed using JASCO J-815 spectrometer (290–650 nm, 25 ± 1 °C). A stock solution of **6** ($2.00 \times 10^{-5}$ M, 5% (v/v) DMSO/CH$_2$Cl$_2$, water content (v/v) 0.04–0.06%) was prepared. Using this solution as a solvent, the stock solution of me-β-D-gal ($3.00 \times 10^{-3}$ M) or me-β-D-glc ($4.00 \times 10^{-3}$ M) were prepared. Aliquots of the guest solutions were added to a UV cell containing the solution of **6** (2.00 mL). UV-visible spectra were recorded, and the association constants (K, M$^{-3}$) were determined using ReactLab software[51].

### ITC experiments

Stock solutions of **6** (0.10–1.00 mM) and guests (4.00–15.0 mM) were prepared separately in 5% (v/v) DMSO/CH$_2$Cl$_2$ (containing 0.04–0.06% water). ITC experiments were conducted by adding the solution of **6** (1.6 mL) to the ITC sample cell, followed by adding each solution of guest using a syringe. Heats of dilution which was obtained by titrating each guest into the ITC sample cell in the absence of **6** were subtracted.

ITC experiments were recorded using MicroCal VP-ITC (spacing time: 240 sec, temperature: $22 \pm 1$ °C, injection volume: 3 or 4 μL). Thermodynamic values were determined using HypCal software[53,54].

## Data collection and structure determination

Crystal structures were solved by the intrinsic phasing method with SHELXT (Ver. 2018/2)[58] and refined by full-matrix least-squares on $F^2$ using SHELXL (Ver. 2018/3)[59] in the Olex2 (Ver. 1.3)[60] program package. All non-hydrogen atoms were refined using anisotropic displacement coefficients. All hydrogen atoms were included in structure factor calculations at idealised positions and were allowed to ride on neighbouring atoms with relative isotropic displacement coefficients.

**4⊃α-D-GF•H₂O:** Diffraction data were acquired at 296 K using a PHOTON 100 CMOS Detector equipped with a Cu-Kα source at the Western Seoul Center of the Korea Basic Science Institute (KBSI). The crystal belonged to the $P2_1$ space group with unit cell parameters: $a = 23.3393(8)$ Å, $b = 15.6893(5)$ Å, $c = 23.6427(8)$ Å and two molecules per unit cell (Z = 2). The indolocarbazole and naphthyridine units were observed to be disordered and modeled using DFIX, SIMU, DELU, and ISOR restraints.

**(6-MP)₂⊃(me-β-D-glc)₂:** Diffraction data were acquired with synchrotron radiation (λ = 0.700000 Å) using a silicon(111) double-crystal monochromator and Rayonix MX225HS detector at 100 K on the BL2D SMC beamline at the Pohang Accelerator Laboratory, Korea. The PAL BL2D-SMDC program[61] was used to collect data and HKL3000sm (Ver. 716.7)[62] was used for cell refinement, reduction and absorption correction. The crystal belonged to the $P2_1$ space group with unit cell parameters of $a = 25.586(5)$ Å, $b = 20.521(4)$ Å, $c = 37.912(8)$ Å and four molecules per unit cell (Z = 4). The indolocarbazole, and naphthyridine units were modelled using DFIX, SIMU, DELU, and ISOR restraints. The 1,2-dichloroethane and two pentanes were refined using isotropic displacement parameters.

**(6-MM)₂⊃(me-β-D-gal•2H₂O)₂ • (6-PP)₂⊃(me-β-L-gal•2H₂O)₂:** Diffraction data were acquired with synchrotron radiation (λ = 0.700000 Å) using a silicon(111) double-crystal monochromator and Rayonix MX225HS detector at 100 K on the BL2D SMC beamline at the Pohang Accelerator Laboratory, Korea. The PAL BL2D-SMDC program[61] was used to collect data, and HKL3000sm (Ver. 716.7)[62] was used for cell refinement, reduction and absorption correction. The crystal belonged to the $P112_1/n$ space group with unit cell parameters: $a = 19.029(4)$ Å, $b = 30.708(6)$ Å, $c = 71.364(14)$ Å and eight molecules per unit cell (Z = 8). Four t-butyl groups were observed to be disordered and modelled using DFIX, SADI and ISOR restraints.

## Data availability

All data are available within the manuscript and supplementary files, or available from the corresponding authors on request. Crystallographic data for the structures reported in this paper have been deposited at the Cambridge Crystallographic Data Centre, under the deposition numbers 2287025 (4⊃α-D-GF • H₂O), 2287027 ((6-MP)₂⊃(me-β-D-glc)₂), and 2287026 ((6-MM)₂⊃(me-β-D-gal•2H₂O)₂ • (6-PP)₂⊃(me-β-L-gal•2H₂O)₂). Copies of these data can be obtained free of charge via www.ccdc.cam.ac.uk/.

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

## Acknowledgements

This study was supported by a National Research Foundation of Korea (NRF) grant funded by the Korean Government (MSIT) (2021R1A2C1093591 to K.-S.J.). The authors acknowledge the Pohang Accelerator Laboratory (PAL) for beamline use (2021-3rd-2D-029, 2022-3rd-2D-007). We would like to thank Editage (www.editage.co.kr) for English language editing.

## Author contributions

G.S. performed all experiments and data analysis. S.L. and G.S. collected X-ray data and refined the structures. K.-S.J. conceived and supervised the project. All authors discussed the results and write, read, and reviewed the manuscript.

## Competing interests

The authors declare no competing interests.
