## [Peer Review File · Nature Communications]

REVIEWER COMMENTS

Reviewer #1 (Remarks to the Author):

This manuscript by Song et al. describes the characterization of complexes in terms of structure and stability between linear diimines with indolocarbazole subunits and monosaccharides. The authors show that a diamine containing a central 1,3-diaminobenzene moiety binds galactose in 2 vol% DMSO/DCM as a 1:1 complex. Interestingly, of the five D-galactose isomers, only the alpha-D-furanose form is bound. Corresponding linear oligomers with 9H-fluorene-2,7-diamine as the central unit also bind monosaccharides, in this case forming 2:2 complexes. The structures of these complexes differ for a glucose-derived and a galactose-derived guest in the helical senses of the receptor subunits. Furthermore, the formation of one complex is entropically favored, while the formation of the other is entropically disfavored, giving rise to an interesting temperature dependence of substrate selectivity.

Overall, the work described in this manuscript is very interesting and provides deep insight into the behavior of an intriguing class of receptors. The apparent high selectivity is unusual in the field of carbohydrate recognition and the temperature dependence of receptor selectivity is unique. Publication of the manuscript in Nature Communication can therefore be recommended. The authors may wish to consider the following suggestions:

- The reference to dynamic combinatorial chemistry in the introduction is probably not very appropriate, because receptor design or the observed template effect does not involve a strong "combinatorial" element.
- Similarly, the concept that seems to underlie receptor design is not entirely credible. The interplay of preorganization and induced fit has been widely studied in supramolecular chemistry since the early days. Well preorganized receptors are difficult to design, as the authors correctly point out, but sacrificing preorganization for flexibility is often not an option either, as it is usually associated with a loss of affinity and/or selectivity. The high selectivity of the oligomers described in this manuscript is certainly striking, but the underlying concept is far from novel. Moreover, substrates are necessary to select an appropriate receptor conformation, and this conformational stabilization is immediately lost upon decomplexation.
- The authors use the term foldamer-based receptor in the introduction, but a foldamer is an oligomer that adopts a folded conformation even in the absence of a guest molecule (or template). What is known about the conformation of free receptors or helical receptor subunits?
- The appearance of sharp signals in the NMR spectra upon addition of D-galactose to a mixture of 1 and 2 is certainly a strong indication of a template effect or the stabilization of a defined receptor conformation by the substrate. The same is true for the formation of receptor 6 in the presence of galactose and glucose. However, the HPLC traces in the Supporting Information indicate that other carbohydrates, which lead to broad and difficult to interpret NMR spectra, also shift the product distribution toward the linear diimines. Thus, other carbohydrates also interact with the receptors, although their interaction is probably less defined. Is it possible to obtain information about these

complexes, e.g. stability constants (by ITC?), to see whether the corresponding complexes are really less stable or only structurally less well defined?

- Finally, the selectivity of the receptors for certain monosaccharides is not rationalized in the manuscript. Is it possible to derive structural information from the crystal structures that would explain why certain monosaccharides probably bind better than others?

Reviewer #2 (Remarks to the Author):

The authors report here the development of foldamer-based receptors for carbohydrate recognition, which can be obtained by complexation-driven equilibrium shifting of imine bonds using monosaccharides as templates. D-galactose drives to the quantitative formation of the first foldamer, which in turn shifts the equilibrium of D-galactose's isomers towards the single stereoisomer α -D-galactofuranose for optimal complementarity. The second foldamer assembles in a dimeric receptor differently depending on whether it is templated with methyl- β -glucopyranoside or methyl- β -galactopyranoside, quantitatively giving rise to different folding structures with two domain-swapping cavities binding two molecules of the guest. Interestingly because of the different thermodynamic origin of the binding of the two glycosides, the foldamer shows a temperature-controlled selection of the guest by reorganizing the binding cavity through unfolding and refolding processes.

Template synthesis by dynamic covalent chemistry, which has already been used for the development of macrocyclic receptors for carbohydrates, is used here to obtain foldamer structures in which adaptive folding allows for the obtainment of dimeric receptors hardly conceivable by rational design. The binding results, although achieved in a low competitive organic solvent, show a high selectivity among the set of investigated carbohydrates and are therefore of relevance to the field of molecular recognition of carbohydrates.

The characterization of the carbohydrate-foldamer complexes has been thoroughly carried out using various calorimetric and spectroscopic techniques including ITC, NMR, circular dichroism, and single crystal X-ray diffraction. The methodology is sound and well described, the results are convincing and support the conclusions, and the authors' interpretation is consistent with the results.

My overall assessment is that the paper is suitable for publication in Nature Communications, provided that the following minor issues are addressed:

- The abbreviation "glu" for glucose can be misleading and is better replaced with the more commonly used "glc".
- A scheme of the atomic labelling of α -D-galactofuranose may be useful for the interpretation of Figure 2e.
- In the title of Table 1 it is more correct to refer to "titration results" rather than "titration data". Furthermore, to avoid misinterpretation of the binding results due to the adimensionality of $\log K$, it is necessary to refer the binding results to the 2:2 complexes in the caption.

- As molecular recognition of carbohydrates is a central topic of the manuscript, the authors need to integrate the references with some reviews on the subject.
- To support the relevance of the measured affinities, authors are encouraged to include references from the current literature concerning the molecular recognition of glucose and galactose by synthetic receptors, both in organic and aqueous media.

Reviewer #3 (Remarks to the Author):

In this article Jeong et al describe the synthesis of new foldamers. Their formations with and without carbohydrates guests was studied. A dual complexation induced equilibrium shift was observed for both the receptor and the guest. This work is interesting, nice X-ray structures of various foldamers, with a carbohydrate guest trapped inside, were obtained and the complexation in solution is supported by a set of complete experiments. Furthermore, at the end an original temperature-controlled selection of bound guests is reported. Thus, this article is well-written and nice and very interesting results have been obtained, thus, it could be published in Nature Communications, but there are issues that need to be solved before publication.

Major points:

- Figure 2, I wonder if for the host 4 alone, the equilibrium is reached. Do the authors follow this procedure: "after 24 h heating at 39 °C" even for the host alone ? This is also surprising that in the SI (Figure S36) only the receptor 4 seems present (without guest). It is not so clear if it is a complexation-driven equilibrium shift. This even less clear for host 6: this host seems to be formed quantitatively without guest (Figure 5 (a) and S37). But maybe it is only because of a change in the solvent used for Figures S36 and S37?

- Figure 2, for glucose and fructose, I wonder if these complex spectra are not due to the complexation of different isomers of these guests, with no clear selectivity, hence a complex spectrum is observed. Numerous equilibrium are probably involved in these cases, as well as the complexation of the various guest isomers in the P and M host, leading to numerous NMR signals. This should be mentioned and if possible demonstrated or ruled out by the authors (maybe by mass ?). Maybe the difference in the spectra is more related to a difference of selectivity depending on the guest: high selectivity: simple NMR, low selectivity: complex NMR. (same comment for host 6)

- Could the authors performed an NMR of 4 in the presence of a mixture of various guests (as in Figure 6 for host 6)

- For Host 4, concerning the determination of the binding constant, numerous equilibrium are involved, I m not sure that K can be obtained by this kind of study, without taking into account all the equilibria. For Host 6, the system seems more simple and the K are probably correct.

- Concerning the temperature-controlled selection of bound guests, could the author confirm that no precipitation occurs at -20°C

Minor points:

- Figure 1 and eq. are very simple, they might be in SI

- Some references are missing, at least this one should be added : Nature Chemistry, 2015, 7, 334

RESPONSE TO REVIEWERS

Reviewer 1

This manuscript by Song et al. describes the characterization of complexes in terms of structure and stability between linear diimines with indolocarbazole subunits and monosaccharides. The authors show that a diamine containing a central 1,3-diaminobenzene moiety binds galactose in 2 vol% DMSO/DCM as a 1:1 complex. Interestingly, of the five D-galactose isomers, only the alpha-D-furanose form is bound. Corresponding linear oligomers with 9H-fluorene-2,7-diamine as the central unit also bind monosaccharides, in this case forming 2:2 complexes. The structures of these complexes differ for a glucose-derived and a galactose-derived guest in the helical senses of the receptor subunits. Furthermore, the formation of one complex is entropically favored, while the formation of the other is entropically disfavored, giving rise to an interesting temperature dependence of substrate selectivity.

Overall, the work described in this manuscript is very interesting and provides deep insight into the behavior of an intriguing class of receptors. The apparent high selectivity is unusual in the field of carbohydrate recognition and the temperature dependence of receptor selectivity is unique. Publication of the manuscript in Nature Communication can therefore be recommended. The authors may wish to consider the following suggestions:

Q1. *The reference to dynamic combinatorial chemistry in the introduction is probably not very appropriate, because receptor design or the observed template effect does not involve a strong "combinatorial" element. Similarly, the concept that seems to underlie receptor design is not entirely credible. The interplay of preorganization and induced fit has been widely studied in supramolecular chemistry since the early days. Well preorganized receptors are difficult to design, as the authors correctly point out, but sacrificing preorganization for flexibility is often not an option either, as it is usually associated with a loss of affinity and/or selectivity. The high selectivity of the oligomers described in this manuscript is certainly striking, but the underlying concept is far from novel. Moreover, substrates are necessary to select an appropriate receptor conformation, and this conformational stabilization is immediately lost upon decomplexation.*

Response: We agree with the comments from reviewer 1. Accordingly, we have revised a section of the introduction and have added appropriate references as below.

(In manuscript, page 1)

“Often, a certain degree of flexibility is inevitable to facilitate an induced fit³ between interacting partners, which may potentially result in decreased affinities and selectivities. The optimisation of binding properties can be achieved through iterative modification and evaluation processes. Another strategy involves complexation-driven equilibrium shifting⁴⁻⁷ to assemble specific receptors that complement to the added guests, utilising dynamic covalent bonds under reversible conditions⁸⁻¹¹. In this method, receptors can be assembled effectively or quantitatively only when tightly binding guest are present together in reaction mixtures.”

(Reference 8, 10, 11)

8. Rowan, S. J., Cantrill, S. J., Cousins, G. R. L., Sanders, J. K. M. & Stoddart, J. F. Dynamic Covalent Chemistry. *Angew. Chem. Int. Ed.* **41**, 898–952 (2002)
10. Jin, Y., Yu, C., Denman, R. J. & Zhang, W. Recent advances in dynamic covalent chemistry. *Chem. Soc. Rev.* **42**, 6634–6654 (2013)
11. Montà-González, G., Sancenón, F., Martínez-Mañez, R. & Martí-Centelles, V. Purely Covalent Molecular Cages and Containers for Guest Encapsulation. *Chem. Rev.* **122**, 13636–13708 (2022).

Q2. The authors use the term *foldamer-based receptor* in the introduction, but a *foldamer* is an oligomer that adopts a folded conformation even in the absence of a guest molecule (or template). What is known about the conformation of free receptors or helical receptor subunits?

Response: In recent years, we have demonstrated that indolocarbazole-pyridine (or naphthyridine) oligomers adopt helical conformations in solution and in the solid states. These oligomers consist of two aryl monomers, indolocarbazole and pyridine (or naphthyridine), linked through ethynyl bonds. The main driving force for helical folding is the dipole-dipole interactions between the two repeating monomers through ethynyl bonds, as depicted below. This folding principle was described in the previous works, together with the X-ray structures (*J. Am. Chem. Soc.* **138**, 92–95 (2016), *Org. Lett.* **19**, 5625–5628 (2017), *Angew. Chem. Int. Ed.* **59**, 22475–22479 (2020)).

Figure. (a) Preferred conformations of indolocarbazole-pyridine (or naphthyridine) trimers. X-ray crystal structures of (b) an indolocarbazole-pyridine heptamer (*J. Am. Chem. Soc.* **138**, 92–95 (2016)) and (c) an indolocarbazole-naphthyridine heptamer (*Org. Lett.* **19**, 5625–5628 (2017)).

Q3. The appearance of sharp signals in the NMR spectra upon addition of *D*-galactose to a

mixture of 1 and 2 is certainly a strong indication of a template effect or the stabilization of a defined receptor conformation by the substrate. The same is true for the formation of receptor 6 in the presence of galactose and glucose. However, the HPLC traces in the Supporting Information indicate that other carbohydrates, which lead to broad and difficult to interpret NMR spectra, also shift the product distribution toward the linear diimines. Thus, other carbohydrates also interact with the receptors, although their interaction is probably less defined. Is it possible to obtain information about these complexes, e.g. stability constants (by ITC?), to see whether the corresponding complexes are really less stable or only structurally less well defined?

Response: As pointed out by Reviewer 1, carbohydrates other than D-galactose also led to equilibrium shifting for assembling diimine **4** (Supplementary Table. 1 and Fig. 4). However, the magnitudes of the equilibrium shifting induced by other carbohydrates are much lower, and their binding affinities for diimine **4** are expected to be much smaller than D-galactose. To obtain quantitative information on the binding affinities and interpret the origin of broad ^1H NMR spectra, we conducted two additional experiments.

First, competitive binding experiments were conducted with **4** and mixed monosaccharide guests (D-glucose, D-galactose, D-mannose, D-fructose, and D-arabinose). Three separate solutions, each containing **4** and different combinations of mixed guests (3 equiv. of each guest), were prepared in 5% (v/v) $\text{DMSO-}d_6/\text{CD}_2\text{Cl}_2$ and ^1H NMR spectra were recorded. A well-resolved, sharp ^1H NMR spectrum was observed only when the guest mixtures contained D-galactose, but the two other solutions yielded broad spectra. The ^1H NMR spectrum in this experiment is identical to the one observed from the solution containing D-galactose alone as a guest. This result clearly indicates that D-galactose binds more strongly than any other monosaccharides.

Supplementary Figure 16. ^1H NMR spectra (400 MHz, 25 °C) of **4** (1.0 mM) in the presence of mixed guests (3 equiv. of each guest) in 5% (v/v) $\text{DMSO-}d_6/\text{CD}_2\text{Cl}_2$.

Next, we also performed CD titrations for monosaccharide guests (D-glucose, D-mannose, D-fructose, and D-arabinose) to compare their binding affinities more quantitatively. As summarized in the Table below, the association constants are smaller by more than one order

of magnitude, compared with that of D-galactose. Here, the reported association constants are not intrinsic binding affinities between **4** and a specific isomer of a monosaccharide. Instead, they are the observed association constants (K_{obs} , M^{-1}) calculated without considering the structural isomeric distributions of monosaccharide guests. Details of all CD titrations are provided in Supplementary Table. 6 and Figs. 20–23.

Supplementary Table 6. Observed association constants (K_{obs} , M^{-1})^[a] between **4** and guests in 10% DMSO/ CH_2Cl_2 (containing 0.04–0.06% water)

Entry	Guest	K_{obs} (M^{-1})
1	D-galactose	54000 (± 1700)
2	D-glucose ^[b]	~4000
3	D-mannose	4580 (± 220)
4	D-fructose	3020 (± 240)
5	D-arabinose	2490 (± 50)

[a] The observed association constants were estimated by nonlinear squares fitting, assuming 1:1 binding modes.

[b] Induced CD signal changes upon binding were too small to accurately calculate the association constant. All titrations were duplicated at 25 ± 1 °C.

(For all CD spectral changes and titration curves, see “Supplementary Information, pp 25–26”).

Similarly, we performed the same additional experiments for diimine **6** to obtain quantitative information on the binding affinities and interpret the origin of broad ^1H NMR spectra. Competitive binding experiments were carried out with **6** and mixed methyl glycoside guests, including methyl β -D-glucopyranoside (me- β -D-glc), methyl α -D-galactopyranoside (me- α -D-gal), methyl β -D-galactopyranoside (me- β -D-gal), methyl α -D-mannopyranoside (me- α -D-man), methyl β -D-xylopyranoside (me- β -D-xyl). Five separate solutions, each containing **6** and different combinations of mixed guests (2.5 equiv. of each guest), were prepared in 5% (v/v) DMSO- d_6 / CD_2Cl_2 and ^1H NMR spectra were recorded. Well-resolved, sharp ^1H NMR spectra were observed only when the guest mixtures contained methyl β -D-glucopyranoside and/or methyl β -D-galactopyranoside, but other combinations of the mixed guests resulted in broad spectra. These findings confirm that both methyl β -D-glucopyranoside and methyl β -D-galactopyranoside bind more strongly than any other methyl glycosides.

Supplementary Figure 17. ^1H NMR spectra (400 MHz, 25 °C) of **6** (1.0 mM) in the presence of mixed guests (2.5 equiv. of each guest) in 5% (v/v) DMSO- d_6 /CD $_2$ Cl $_2$.

We also conducted isothermal titration calorimetry (ITC) experiments to determine the association constants between **6** and some methyl glycoside guests. The association constants were calculated, assuming 2:2 receptor/guest binding modes, as for me- β -D-glc and me- β -D-gal. As summarized in the table below, the association constants ($\log K$ for 2:2 complex) for me- β -D-xyl, me- α -D-gal, and me- α -D-man are noticeably smaller compared to those of me- β -D-glc and me- β -D-gal under identical conditions. All ITC results and binding isotherms are shown in Supplementary Table 7 and Figs. 29–33.

Supplementary Table 7. Titration results between **6** and methyl glycoside guests in 5% DMSO/CH $_2$ Cl $_2$ (containing 0.04–0.06% water)

Entry	Guest	Log K (2:2 complex)	ΔG°	ΔH°	$T\Delta S^\circ$
			(kJ mol $^{-1}$)		
1	me- β -D-glc	13.5 (± 0.1)	-76.4	-56.0	+20.4
2	me- β -D-gal	13.2 (± 0.1)	-74.4	-130.2	-55.8
3	me- β -D-xyl	10.1 (± 0.2)	-57.2	-38.9	+18.3
4	me- α -D-gal	10.1 (± 0.1)	-57.2	-42.2	+15.0
5	me- α -D-man	9.9 (± 0.2)	-56.0	-24.7	+31.3

$K = [\text{diimine}_2 \cdot \text{guest}_2] / [\text{diimine}]^2 [\text{guest}]^2$. All titrations were duplicated at 22 ± 1 °C

(For all ITC experimental measurements, see “Supplementary Information, pp 30-34”).

(In manuscript, page 6)

“Selective binding of **4** with α -D-GF was confirmed by competitive binding experiments using mixed monosaccharide guests including D-glucose, D-galactose, D-mannose, D-fructose, and D-arabinose. Three separate solutions were prepared, each containing **4** and different combinations of mixed guests (3 equiv of each guest) in 5% (v/v) DMSO- d_6 /CD₂Cl₂. A well-resolved, sharp ¹H NMR spectrum was exclusively observed in the solution containing D-galactose among the mixed guests. Notably, this spectrum was identical to the one obtained for the complex between **4** and D-galactose as a single guest (Fig 1f). In contrast, the other solutions lacking D-galactose in the mixed guests yielded broad spectra (Supplementary Fig. 16). These results indicate that α -D-GF selectively and strongly binds to **4**, enabling the quantitative assembly of **4** under the given conditions. For comparison, CD titrations for D-glucose, D-mannose, D-fructose, and D-arabinose were also performed, and the observed association constants were estimated based on 1:1 binding modes. As anticipated, all these values are smaller by more than one order of magnitude compared with that ($K_{\text{obs}} = 5.40 \times 10^4 \text{ M}^{-1}$) of D-galactose (Supplementary Table. 6 and Figs. 21–23).”

(In manuscript, page 15)

“To further demonstrate the selective binding of **6** towards me- β -D-glc and me- β -D-gal, we conducted competitive binding experiments. Five separate solutions were prepared in 5% (v/v) DMSO- d_6 /CD₂Cl₂, each containing **6** and different combinations of mixed glycoside guests (me- β -D-glc, me- α -D-gal, me- β -D-gal, me- α -D-man, and me- β -D-xyl). Well-resolved ¹H NMR spectra were observed only when the guest mixtures contained me- β -D-glc and/or me- β -D-gal, while other combinations of the guests resulted in broad spectra (Supplementary Fig. 17). These results strongly suggest that both me- β -D-glc and me- β -D-gal bind more strongly to **6** than other methyl glycosides under the given conditions. As anticipated, isothermal titration calorimetry (ITC) experiments demonstrated that the association constants of **6** for me- α -D-gal, me- α -D-man, and me- β -D-xyl are much smaller than those of me- β -D-glc and me- β -D-gal (Supplementary Table. 7 and Figs. 31–33).”

Q4. *Finally, the selectivity of the receptors for certain monosaccharides is not rationalized in the manuscript. Is it possible to derive structural information from the crystal structures that would explain why certain monosaccharides probably bind better than others?*

Response: As suggested by Reviewer 1, we performed more structural analyses to rationalize the binding selectivity of **4**, based on their crystal structure and computer models (MacroModel 9.1, MMFFs force field).

In the crystal structure of **4** \supset α -D-GF \cdot H₂O, we identified 14 hydrogen bonds between **4**, α -D-GF and a water molecule, as described in Fig. 2e and Supplementary Table. 14. When substituting α -D-GF with β -D-GF in a computer model using this crystal structure, the two hydrogen bonds initially formed with the anomeric OH are disrupted (Supplementary Fig. 41b). In the case of the pyranose isomers (Supplementary Figs. 41 c and 41d), the hydrogen-bonding array of **4** \supset α -D-GF \cdot H₂O are significantly perturbed, resulting in even fewer

hydrogen bonds between **4** and the pyranose isomers.

Regarding diimine **6**, it has been found to form two distinct 2:2 complexes with *me*- β -D-glc and *me*- β -D-gal. Notably, the folding conformations of **6** in these two crystal structures exhibit considerable variability and adaptability. The conformational diversity and adaptability pose great challenges in explaining the origin of binding selectivity at the atomic level, based on simple computer models. Instead, we opted to determine and compare the binding affinities for analogous methyl glycosides to provide quantitative insights into binding selectivity as described in the previous part.

(In Supplementary Information)

Supplementary Figure 41. Tube representations of (a) the X-ray crystal structure $4 \supset \alpha\text{-D-GF} \cdot \text{H}_2\text{O}$, and (b-d) energy minimized molecular models depicting diimine **4** complexes with three other isomers of D-galactose: (b) $\beta\text{-D-GF}$, (c) $\alpha\text{-D-GP}$, and (d) $\beta\text{-D-GP}$, based on the crystal structure. The purple- and green-colored dashed lines represent disrupted hydrogen bonds upon isomerization.

(In manuscript, page 6)

“Computer modeling^{46,47}, utilising this crystal structure, indicates that the two hydrogen bonds, initially formed with the anomeric OH, are disrupted when substituting $\alpha\text{-D-GF}$ with $\beta\text{-D-GF}$. In the case of the pyranose isomers, the hydrogen-bonding array of $4 \supset \alpha\text{-D-GF} \cdot \text{H}_2\text{O}$ are significantly perturbed, resulting in even fewer hydrogen bonds between **4** and the pyranose isomers (Supplementary Fig. 41).”

(Reference 46, 47)

46. Mohamadi, F. et al. MacroModel—an integrated software system for modeling organic and bioorganic molecules using molecular mechanics. *J. Comput. Chem.* **11**, 440–467 (1990).

47. MacroModel, version 9.1; Schrödinger, LLC: New York, 2005.

Reviewer 2

The authors report here the development of foldamer-based receptors for carbohydrate recognition, which can be obtained by complexation-driven equilibrium shifting of imine bonds using monosaccharides as templates. D-galactose drives to the quantitative formation of the first foldamer, which in turn shifts the equilibrium of D-galactose's isomers towards the single stereoisomer α -D-galactofuranose for optimal complementarity. The second foldamer assembles in a dimeric receptor differently depending on whether it is templated with methyl β -glucopyranoside or methyl β -galactopyranoside, quantitatively giving rise to different folding structures with two domain-swapping cavities binding two molecules of the guest. Interestingly because of the different thermodynamic origin of the binding of the two glycosides, the foldamer shows a temperature-controlled selection of the guest by reorganizing the binding cavity through unfolding and refolding processes.

Template synthesis by dynamic covalent chemistry, which has already been used for the development of macrocyclic receptors for carbohydrates, is used here to obtain foldamer structures in which adaptive folding allows for the obtainment of dimeric receptors hardly conceivable by rational design. The binding results, although achieved in a low competitive organic solvent, show a high selectivity among the set of investigated carbohydrates and are therefore of relevance to the field of molecular recognition of carbohydrates.

The characterization of the carbohydrate-foldamer complexes has been thoroughly carried out using various calorimetric and spectroscopic techniques including ITC, NMR, circular dichroism, and single crystal X-ray diffraction. The methodology is sound and well described, the results are convincing and support the conclusions, and the authors' interpretation is consistent with the results.

My overall assessment is that the paper is suitable for publication in Nature Communications, provided that the following minor issues are addressed:

Q1. *The abbreviation “glu” for glucose can be misleading and is better replaced with the more commonly used “glc”.*

Response: The abbreviation “glu” has been changed to “glc”.

Q2. *A scheme of the atomic labelling of α -D-galactofuranose may be useful for the interpretation of Figure 2e.*

Response: Atomic labelling of D-galactofuranose has been added in Fig. 1d for the interpretation of Fig 2e (original) / Fig 1f (revised).

Q3. *In the title of Table 1 it is more correct to refer to “titration results” rather than*

“titration data”. Furthermore, to avoid misinterpretation of the binding results due to the adimensionality of $\log K$, it is necessary to refer the binding results to the 2:2 complexes in the caption.

Response: In the title of Table 1, term “titration data” has been changed to “titration results”. Additionally, “2:2 complex” has been added next to $\log K$ in Table 1 and “ $K = [\text{diimine}_2 \cdot \text{guest}_2] / [\text{diimine}]^2 [\text{guest}]^2$ ” in the caption below Table 1.

Q4. As molecular recognition of carbohydrates is a central topic of the manuscript, the authors need to integrate the references with some reviews on the subject. To support the relevance of the measured affinities, authors are encouraged to include references from the current literature concerning the molecular recognition of glucose and galactose by synthetic receptors, both in organic and aqueous media.

Response: As commented by reviewer 2, we have addressed the development of synthetic receptors for carbohydrates and provided appropriate references in the manuscript, as shown below.

(In manuscript, page 3)

“In this study, we selected monosaccharides as guests that have multiple hydroxyl groups capable of forming hydrogen bonds with the imine product. Synthetic receptors for carbohydrates have been widely studied^{16,29–39,43}, but achieving selective binding of specific carbohydrates remains a great challenge due to the structural and functional group similarity.”

(For reviews, see Reference 29–32)

29. Mazik, M. Molecular recognition of carbohydrates by acyclic receptors employing noncovalent interactions. *Chem. Soc. Rev.* **38**, 935–956 (2009).
30. Arnaud, J., Audfray, A. & Imberty, A. Binding sugars: from natural lectins to synthetic receptors and engineered neolectins. *Chem. Soc. Rev.* **42**, 4798–4813 (2013).
31. Sun, X. & James, T. D. Glucose Sensing in Supramolecular Chemistry. *Chem. Rev.* **115**, 8001–8037 (2015).
32. Davis, A. P. Biomimetic carbohydrate recognition. *Chem. Soc. Rev.* **49**, 2531–2545 (2020).

(For synthetic receptors capable of binding carbohydrates, see Reference 33–38)

33. Nativi, C. et al. Pyrrolic Tripodal Receptors Effectively Recognizing Monosaccharides. Affinity Assessment through a Generalized Binding Descriptor. *J. Am. Chem. Soc.* **129**, 4377–4385 (2007).
34. Rózs, P. et al. Enantioselective carbohydrate recognition by synthetic lectins in water. *Chem. Sci.* **8**, 4056–4061 (2017).
35. Mateus, P., Wicher, B., Ferrand, Y. & Huc, I. Carbohydrate binding through first- and second-sphere coordination within aromatic oligoamide metallofoldamers. *Chem. Commun.* **54**, 5078–5081 (2018).

36. Tromans, R. A. et al. A biomimetic receptor for glucose. *Nat. Chem.* **11**, 52–56 (2019).
37. Mateus, P., Chandramouli, N., Mackereth, C. D., Kauffmann, B., Ferrand, Y. & Huc, I. Allosteric Recognition of Homomeric and Heteromeric Pairs of Monosaccharides by a Foldamer Capsule. *Angew. Chem. Int. Ed.* **59**, 5797–5805 (2020).
38. Timmer, B. J. J., Kooijman, A., Schaapkens, X. & Mooibroek, T. J. A Synthetic Galectin Mimic. *Angew. Chem. Int. Ed.* **60**, 16178–16183 (2021).

Reviewer 3

In this article Jeong et al describe the synthesis of new foldamers. Their formations with and without carbohydrates guests was studied. A dual complexation induced equilibrium shift was observed for both the receptor and the guest. This work is interesting, nice X-ray structures of various foldamers, with a carbohydrate guest trapped inside, were obtained and the complexation in solution is supported by a set of complete experiments. Furthermore, at the end an original temperature-controlled selection of bound guests is reported. Thus, this article is well-written and nice and very interesting results have been obtained, thus, it could be published in Nature Communications, but there are issues that need to be solved before publication.

Q1. Figure 2, I wonder if for the host **4** alone, the equilibrium is reached. Do the authors follow this procedure: “after 24 h heating at 39 °C” even for the host alone?

Response: In Fig 2e (original) / Fig. 1f (revised), the bottom ¹H NMR spectrum is the one obtained after 24 h heating at 39 °C of a mixture containing **1** (2 equiv.), benzene-1,3-diamine (**2**) (1 equiv.), chloroacetic acid (0.2 equiv.) in 2% (v/v) DMSO-*d*₆/CD₂Cl₂ (containing ~0.06% v/v water) without any guest. The ratios of tetramer (**1**), monoimine (**3**) and diimine (**4**) were determined to be approximately 29:28:43. To confirm whether the equilibrium is reached under the given conditions, diimine (**4**) (1 equiv.) and chloroacetic acid (0.2 equiv.) were dissolved in 2% (v/v) DMSO-*d*₆/CD₂Cl₂ (containing ~0.06% v/v water) and was allowed to stand at 39 °C for 24 h. The ratios of tetramer (**1**), monoimine (**3**) and diimine (**4**) were determined to be ~ 27:26:47 (Figure b), which are essentially equal to those calculated in Figure a. This result confirms that the equilibrium has been established under the given conditions.

Figure. Partial ¹H NMR spectra (400 MHz, 25 °C) after heating for 24 h at 39 °C of each solution (2% DMSO-*d*₆/CD₂Cl₂ (water ~0.06 v/v %) containing (a) **1** (2 equiv.), benzene-1,3-diamine (**2**) (1 equiv.) and chloroacetic acid (0.2 equiv.), as well as (b) pure diimine (**4**) (1 equiv.) and chloroacetic acid (0.2 equiv.).

Q2. *This is also surprising that in the SI (Figure S36) only the receptor 4 seems present (without guest). It is not so clear if it is a complexation-driven equilibrium shift. This even less clear for host 6: this host seems to be formed quantitatively without guest (Figure 5 (a) and S37). But maybe it is only because of a change in the solvent used for Figures S36 and S37?*

Response: As described in the Methods section, diimines **4** (97 % isolated yield) and **6** (95 % isolated yield) were synthesized in the presence of D-galactose and methyl β -D-glucopyranoside (or methyl β -D-galactopyranoside), respectively. These isolated diimines were used for both binding studies (Fig 5a (original) / Fig 4a (revised)) and also for spectroscopic characterization (^1H & ^{13}C NMR spectra in Supplementary Figs. 36 and 37 (original) / Supplementary Figs. 45 and 46 (revised)).

Q3. *Figure 2, for glucose and fructose, I wonder if these complex spectra are not due to the complexation of different isomers of these guests, with no clear selectivity, hence a complex spectrum is observed. Numerous equilibrium are probably involved in these cases, as well as the complexation of the various guest isomers in the P and M host, leading to numerous NMR signals. This should be mentioned and if possible demonstrated or ruled out by the authors (maybe by mass?). Maybe the difference in the spectra is more related to a difference of selectivity depending on the guest: high selectivity: simple NMR, low selectivity: complex NMR. (same comment for host 6).*

Response: Yes, we agree with Reviewer 3. The broad, unresolved ^1H NMR spectra observed for guests other than D-galactose for **4** and me- β -D-glc (or me- β -D-gal) for **6** arise from multiple equilibria between the unbound/its complexes, as well as between the complexes of possible different isomers for **4** under the given conditions. As detailed in the **Response to Reviewer 1, Q3**, we conducted two additional experiments, competitive binding experiments and CD titrations (or ITC experiments), to clarify these issues. These experiments indicate that broad, unresolved ^1H NMR spectra observed for other guests are primarily due to the lower binding affinities under the given conditions.

Q4. *Could the authors perform an NMR of 4 in the presence of a mixture of various guests (as in Figure 6 for host 6)*

Response: Please refer to the **Response to Reviewer 1, Q3**. As suggested by Reviewer 3, we recorded a ^1H NMR spectrum of diimine **4** in the presence of mixed guests (D-glucose, D-galactose, D-mannose, D-fructose, and D-arabinose). As shown below, a well-resolved ^1H NMR spectrum corresponding to a complex between diimine **4** and D-galactose can be only seen.

Figure. ^1H NMR spectra (400 MHz, 25 °C) of **4** (1.0 mM) in the presence of a D-galactose (3 equiv.), and mixed guests (3 equiv. of each guest) in 5% (v/v) DMSO- d_6 /CD $_2$ Cl $_2$

Q5. *For Host 4, concerning the determination of the binding constant, numerous equilibrium are involved, I m not sure that K can be obtained by this kind of study, without taking into account all the equilibria. For Host 6, the system seems more simple and the K are probably correct.*

Response: We agree with Reviewer 3. Simple monosaccharides exist in equilibrium mixtures between structural (or constitutional) isomers. The distributions of the isomers depend on the environments and can be also variable during titrations via complexation-induced equilibrium shifts. Taking into account all these equilibria is highly complicated and challenging. In general, the association constants between synthetic receptors and simple monosaccharides have been calculated without considering the structural isomeric distributions of monosaccharide guests. Therefore, the reported association constants are the observed values, not intrinsic ones between **4** and a specific isomer of a monosaccharide. We clarified this issue in the manuscript.

(In manuscript, page 6)

“Non-linear regression analysis⁴⁴ afforded an observed association constant (K_{obs}) of $5.40 (\pm 0.17) \times 10^4 \text{ M}^{-1}$ (Supplementary Fig. 20). It is noted that this value is not the intrinsic binding constant between diimine **4** and α -D-GF but the observed one, calculated without considering the structural isomer distributions of D-galactose.”

Q6. *Concerning the temperature-controlled selection of bound guests, could the author confirm that no precipitation occurs at -20°C*

Response: We carefully examined the ^1H NMR solution used for the temperature-controlled selection of bound guests and confirmed the absence of precipitation at -20°C . Furthermore, ^1H NMR signals for unbound methyl β -D-glucopyranoside were confirmed in the spectrum without any loss signal intensities.

Q7. *Figure 1 and eq. are very simple, they might be in SI*

Response: Figs. 1 and 2 were modified and combined into Fig. 1. Additionally, equation. 1 was removed from the main Text. Instead, it was included as a caption of Table 1 and also in the Supplementary Information.

Q8. *Some references are missing, at least this one should be added : Nature Chemistry, 2015, 7, 334*

Response: The suggested reference (*Nature Chemistry, 2015, 7, 334*) was included in the original version (Reference 16), and two additional relevant references have been added as shown below

(Reference 35 and 37)

35. Mateus, P., Wicher, B., Ferrand, Y. & Huc, I. Carbohydrate binding through first- and second-sphere coordination within aromatic oligoamide metallofoldamers. *Chem. Commun.* **54**, 5078–5081 (2018).
37. Mateus, P., Chandramouli, N., Mackereth, C. D., Kauffmann, B., Ferrand, Y. & Huc, I. Allosteric Recognition of Homomeric and Heteromeric Pairs of Monosaccharides by a Foldamer Capsule. *Angew. Chem. Int. Ed.* **59**, 5797–5805 (2020).

REVIEWERS' COMMENTS

Reviewer #1 (Remarks to the Author):

The authors have revised their manuscript very carefully, taking into account all the reviewers' comments. The additional binding studies and computational work provide valuable additional information that further supports the authors' interpretation. The overall quality of the manuscript has been further improved by the revisions, and publication can therefore be recommended without reservation.

Reviewer #3 (Remarks to the Author):

The authors have given convincing answer to my previous questions. This manucripit is remarkable and of high interest and is now suitable for publications in Nature Communications.